# Stimuli-responsive rotaxane-branched dendronized polymers with tunable thermal and rheological properties

Yu Zhu[1], Hanqiu Jiang [2,3], Weiwei Wu[4], Xiao-Qin Xu[1], Xu-Qing Wang [1]✉, Wei-Jian Li[1], Wei-Tao Xu[1], GengXin Liu[4], Yubin Ke[2,3], Wei Wang [1]✉ & Hai-Bo Yang [1]✉

Aiming at the creation of polymers with attractive dynamic properties, herein, rotaxane-branched dendronized polymers (DPs) with rotaxane-branched dendrons attached onto the polymer chains are proposed. Starting from macromonomers with both rotaxane-branched dendrons and polymerization site, targeted rotaxane-branched DPs are successfully synthesized through ring-opening metathesis polymerization (ROMP). Interestingly, due to the existence of multiple switchable [2]rotaxane branches within the attached dendrons, anion-induced reversible thickness modulation of the resultant rotaxane-branched DPs is achieved, which further lead to tunable thermal and rheological properties, making them attractive platform for the construction of smart polymeric materials.

Since Hermann Staudinger first coined the concept of polymerization in 1920[1], the rapid development of polymer chemistry has been witnessed during the past century and polymers have greatly advanced the progress of society. Relying on the great power of polymer synthesis, diverse polymers with well-tailored architectures and properties have been successfully prepared, laying the foundation for the practical applications of polymers in diverse fields. To further enrich the toolbox of polymers, the design and construction of polymers with desired properties and functions remains an attractive topic in polymer chemistry and materials science[2–5]. In particular, the marriage of traditional polymers with other macromolecules can give rise to functional polymeric materials with attractive properties and promising applications. Dendronized polymers (DPs) are a representative example of such ingenious integration. Seminally reported by the groups of Tomalia, Schlüter, and Percec, DPs have resulted in a new research area at the interfaces of dendrimer chemistry, polymer chemistry, and materials science that has generated intense interest[6–18]. The introduction of branched

dendrons as bulky pendant groups onto a linear polymer chain endows the resultant DPs with not only appealing nanoscale hierarchical architectures but also intriguing properties, making them privileged platforms for practical applications in diverse fields such as drug/gene delivery[19–22], bioimaging[23–25], electronics[26–28], and stimuli-responsive materials[29–32].

As the key structural characteristic of DPs, their thickness could be regarded as a new variable for determining their properties in addition to their chemical structures and chain lengths[33–35]. In particular, the further introduction of stimuli-responsive properties endows the resultant DPs with unique dynamic features, making them particularly attractive for the preparation of smart polymeric materials[36–39]. For instance, by attaching dendritic oligo(ethylene glycols) (OEG) onto polymer chains, a series of DPs with interesting thermoresponsive thicknesses have been successfully constructed, which could be applied as smart delivery vehicles for various guests such as dyes, siRNA, and stem cells, highlighting their great potential for practical applications[40–43]. In addition, dendronized

[1]Shanghai Key Laboratory of Green Chemistry and Chemical Processes, School of Chemistry and Molecular Engineering, East China Normal University, Shanghai 200062, P. R. China. [2]Spallation Neutron Source Science Center, Dongguan 523803, P. R. China. [3]Institute of High Energy Physics, Chinese Academy of Sciences (CAS), Beijing 100049, P. R. China. [4]State Key Laboratory for Modification of Chemical Fibers and Polymer Materials, Center for Advanced Low-dimension Materials, College of Material Science and Engineering, Donghua University, Shanghai 201620, P. R. China. ✉e-mail: xqwang@chem.ecnu.edu.cn; wwang@chem.ecnu.edu.cn; hbyang@chem.ecnu.edu.cn

poly(phenylacetylene)s bearing the second generation lysine dendrons through a urea group has been developed, which could serve as excellent anion receptors for size-selective colorimetric anion sensing such as acetate anion[44]. Thus, the further development of stimuli-responsive DPs with tailored thicknesses, particularly those with new switching mechanisms, is of great importance.

Notably, along with the development of DPs, the rapid development of supramolecular chemistry and mechanically interlocked molecules (MIMs) has also injected new vitality to DPs[45–53]. For instance, Stoddart et al. demonstrated the construction of supramolecular dendronized polyacetylenes (DPAs) through the formation of pseudo[2]rotaxanes as linkers between the polymer backbones and dendrons. More importantly, through an acid/base induced threading-dethreading process, the controllable reversible assembly/disassembly of the resultant supramolecular DPs was successfully achieved, which induced remarkable conformational changes in the polymer backbone[54]. According to this inspiring work, the introduction of rotaxane units into DPs as switchable motifs would endow them with attractive dynamic properties, thus offering a new switching mechanism for the construction of stimuli-responsive DPs. However, in Stoddart's work, the pseudo[2]rotaxane moieties only serve as linkers between the polymer backbone and dendrons. So far, attributed to the difficulties in the synthesis of dendrons with rotaxane units as branches, DPs with rotaxane-branched dendrons have never been reported. Herein, based on our ongoing interests in MIMs, particularly rotaxane-branched dendrimers[55–63], by attaching rotaxane-branched dendrons onto the polymer chains, rotaxane-branched dendronized polymer as a type of DP was proposed and synthesized via the ring-opening metathesis polymerization (ROMP) of macromonomers. More importantly, the existence of multiple switchable [2]rotaxane units within the dendron skeleton endows the resultant rotaxane-branched DPs with attractive stimuli-responsive features. Upon the addition of an external stimulus that triggers the motions of the rotaxane branching points away from the polymer backbone, the stretching of the rotaxane-branched dendrons will result in an increased thickness, thus further leading to tunable thermal and rheological properties (Fig. 1).

## Results

### Design, synthesis, and characterization of rotaxane-branched dendronized polymers

In our study, the macromonomer method[64–66] was employed for the synthesis of targeted rotaxane-branched DPs, thus the macromonomers **MGn** (n = 1, 2, 3) with both the norbornene (NB) moiety as the polymerization site and rotaxane-branched dendrons were first synthesized through a controllable divergent approach. Notably, for **MG1**, although it only carries one rotaxane unit, two growth sites on the pillar[5]arene macrocycle was introduced as a branching point, thus it can be regarded as the first generation macromonomer. To reduce the steric hindrance during polymerization, a flexible alkyl link was inserted between the NB and the rotaxane-branched dendrons as a spacer. Notably, to minimize the possible negative effects of the spacer on the subsequent study on thickness regulation, a relatively short n-hexyl unit was selected. More importantly, [2]rotaxane **1** with a urea moiety as a stimuli-responsive site in the axle component was selected as the key building block for the synthesis of rotaxane-branched dendrons. Upon the addition or removal of acetate anions that can bind with the urea moiety, the pillar[5]arene macrocycle could undergo reversible motions along the axle component between the initial urea moiety and alkyl chain moiety[58]. Such dynamic feature of switchable [2]rotaxane **1** would further endow the targeted rotaxane-branched DPs with anion-induced stimuli responsiveness, therefore making the thickness regulation of the resultant rotaxane-branched DPs possible.

The first-generation macromonomer **MG1** was prepared in 80% yield by CuI-catalyzed coupling reaction between [2]rotaxane **1** and compound **2** with both NB and alkyne moieties as two tails. The sequential deprotection of **MG1** by tetrabutylammonium fluoride trihydrate (Bu₄NF·3H₂O) led to the preparation of **MG1-YNE** with two alkyne termini, which further reacted with [2]rotaxane **1** in the presence of CuI as a catalyst to afford the second-generation macromonomer **MG2** in 71% yield. By repeating such deprotection-coupling reactions, the third-generation macromonomer **MG3** with seven individual [2]rotaxane branches was then successfully prepared. Notably, all the resultant macromonomers were easily purified by flash column

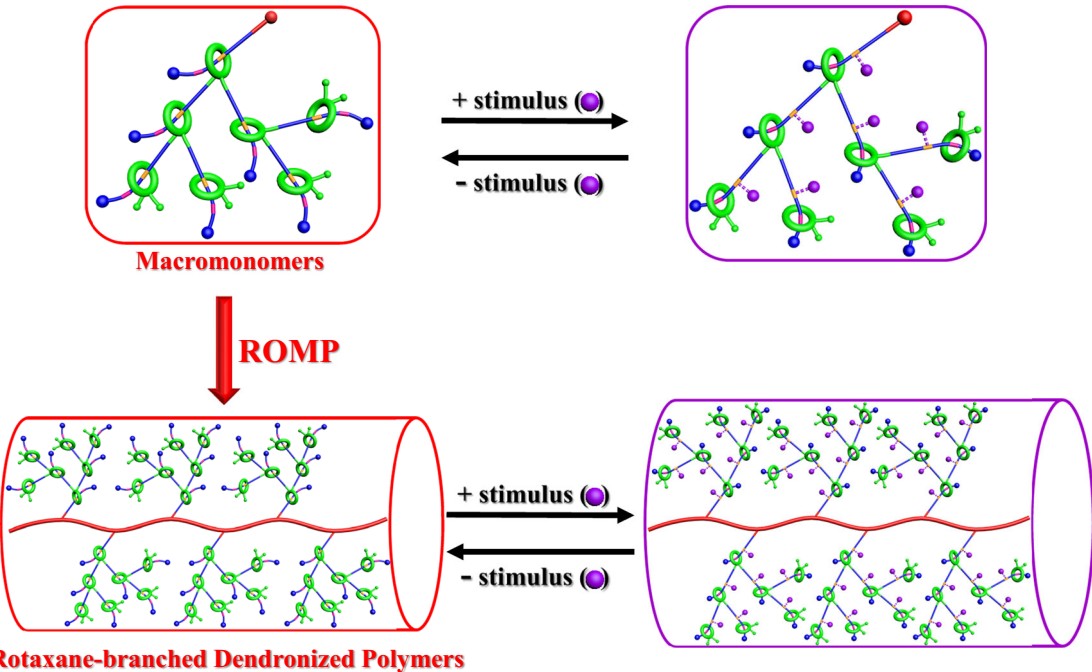

**Fig. 1 | The design strategy of stimuli-responsive rotaxane-branched dendronized polymers.** Schematic illustration of the proposed rotaxane-branched dendronized polymers with tailored thickness through the attachment of switchable rotaxane-branched dendrons onto the polymer chains.

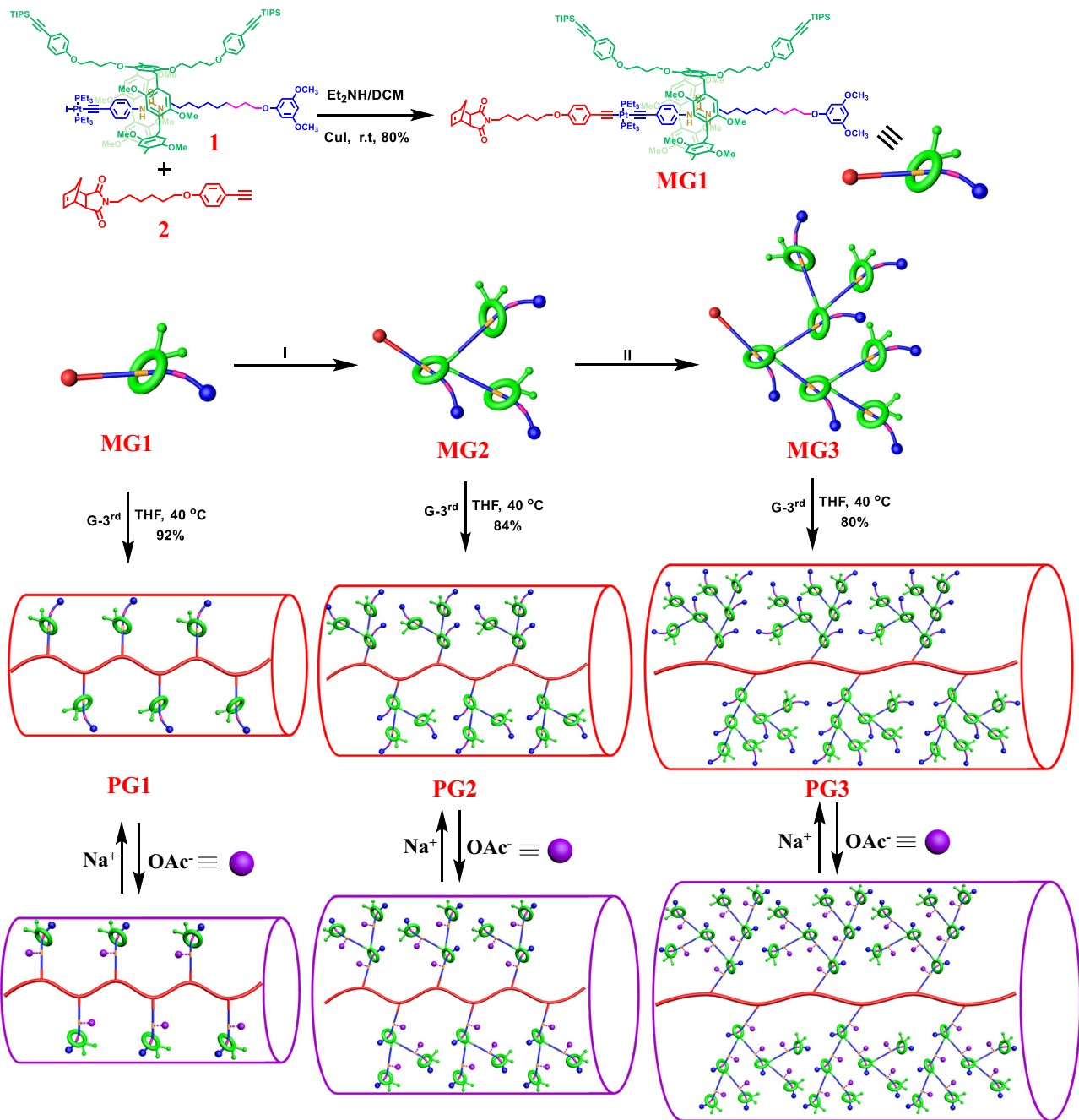

**Fig. 2 | Synthesis of rotaxane-branched DPs PGn and the schematic illustration of their stimuli-responsive features.** Synthesis of macromonomers **MGn** by a controllable divergent strategy and corresponding rotaxane-branched DPs **PGn** through ROMP, and the anion-induced thickness modulation of **PGn** (n = 1, 2, 3).

Reaction conditions: (I) (**a**) Bu₄NF·3H₂O, THF, r.t., 4 h, 91%; (**b**) **1**, CuI, DCM/Et₂NH, r.t., overnight, 78%; (II) (**a**) Bu₄NF·3H₂O, THF, r.t., 4 h, 88%; (**b**) **1**, CuI, DCM/Et₂NH, r.t., overnight, 65%.

chromatography and preparative gel permeation chromatography (GPC) in up to gram scales. In addition, these macromonomers reveal nice solubility and high stability in common solvents such as DCM, chloroform, and THF, making them excellent candidates for further polymerization (Fig. 2).

In the ¹H NMR spectra of the resultant macromonomers **MGn** (n = 1, 2, 3), the proton signal attributed to the terminal alkyne moiety of **2** disappeared, and the signals ascribed to the olefinic protons of NB ring at 6.28 ppm were observed, indicating the successful connection of the polymerization site with rotaxane-branched dendrons through coupling reaction. More importantly, the peaks that are attributed to the protons on the axle of the [2]rotaxane units (particularly those below 0.0 ppm) remained, suggesting that the rotaxane units were

kept intact during the synthetic processes. In addition, remarkable downfield shifts from 8.91 ppm (**1**) to 11.50 ppm (**MG1**), 11.84 ppm (**MG2**), 11.77 ppm (**MG3**) were found in the ³¹P NMR spectra of macromonomers **MGn** (n = 1, 2, 3), suggesting the formation of platinum-acetylide links during the macromonomer growth process. Moreover, as revealed by the MALDI-TOF-MS spectra, peaks of m/z = 2623.3, m/z = 6836.6, m/z = 15260.2 were found, which agreed with the theoretical values of [**MG1** + H]⁺ (m/z = 2624.3), [**MG2** + Li]⁺ (m/z = 6842.6), [**MG3** + H]⁺ (m/z = 15260.1) ions, respectively, further confirming the successful synthesis of these macromonomers (Supplementary Figs. 10–13, 19–22, and 28–31).

With these macromonomers in hand, the synthesis of targeted rotaxane-branched DPs through ROMP was then carried out. Firstly, by

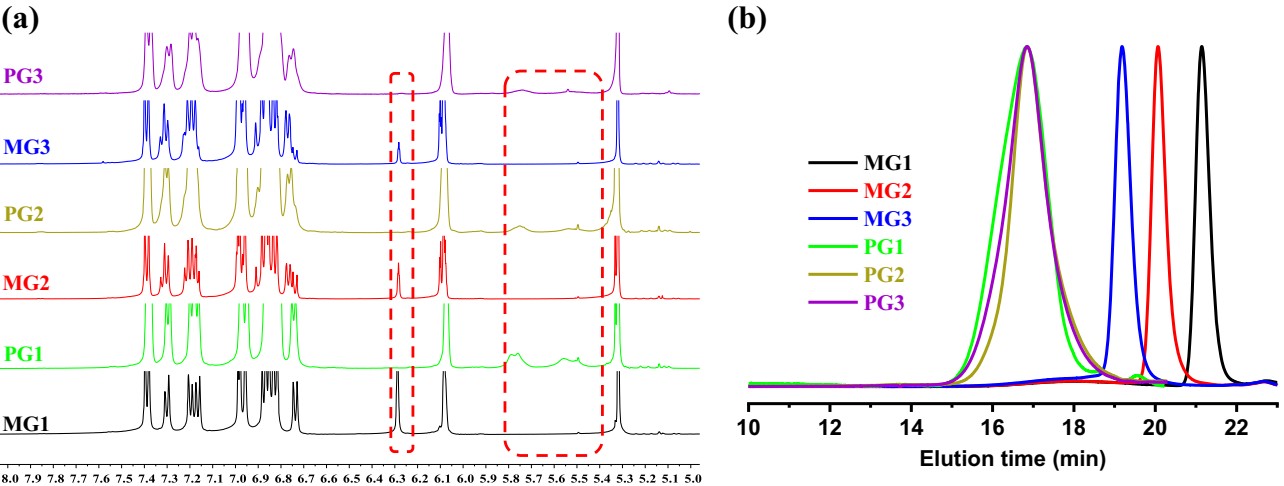

**Fig. 3 | Structural characterization of macromonomers MGn and corresponding rotaxane-branched DPs PGn. a** The partial $^1$H-NMR spectra of macromonomers **MG1**-**MG3** and corresponding rotaxane-branched DPs **PG1**-**PG3**. Full $^1$H-NMR spectra are available in Supplementary Fig. 46. **b** GPC traces of macromonomers **MG1**-**MG3** and corresponding rotaxane-branched DPs **PG1**-**PG3**.

using Grubbs third-generation catalyst as initiator, the rotaxane-branched DP **PG1** was successfully obtained from the corresponding macromonomer **MG1** with the feed ratio of monomer to the initiator ([M]/[I]) of 200:1 (Entry 1, Supplementary Table 1). However, in the case of **MG2** and **MG3**, when the feed ratios were sent to be 200:1, possibly attributed to the enhanced steric hindrances, these macromonomers could not be completely consumed. As determined by TLC monitoring as well as GPC traces (Supplementary Fig. 39), a large amount of unpolymerized macromonomers were found even after prolonging the polymerization times to 48 h. Therefore, the feed ratios were reduced to 100:1 for **MG2** and 50:1 for **MG3**, respectively, which then led to the full conversion of these macromonomers (Supplementary Table 1, Entry 2 and 3). Notably, due to the steric hindrance of the dendrons with rigid pillar[5]arene rings, an elevated temperature (40 °C) was necessary for the initiator activation and complete conversion of the macromonomers. Notably, all the rotaxane-branched DPs could be synthesized on ca. 500 mg scale, which is sufficient for practical use. In the $^1$H NMR spectra of all the resultant rotaxane-branched DPs, the signals attributed to the olefinic protons of NB ring at 6.28 ppm disappeared, and new olefinic proton signals on the polymer backbones at 5.80–5.53 ppm were observed, clearly indicating the formation of desired polymer chains (Fig. 3a). Moreover, the peaks ascribed to the rotaxane units remained, suggesting that the rotaxane-branched dendrons remained intact during the polymerization processes (Supplementary Fig. 46). As determined by multiangle laser light scattering (MALLS) detector, the absolute molecular weights of the resultant rotaxane-branched DPs were 498 kDa for **PG1**, 403 kDa for **PG2**, and 433 kDa for **PG3**, respectively, with acceptable dispersity (1.28 for **PG1**, 1.21 for **PG2**, and 1.23 for **PG3**). These absolute molecular weight values are much higher than that of corresponding macromonomers (2.5 kDa for **MG1**, 6.8 kDa for **MG2**, and 15.2 kDa for **MG3**), again indicating the successful formation of targeted rotaxane-branched DPs (Fig. 3b). As expected, for **PG3**, attributed to the enhanced steric demand for the macromolecule along with the increase in branching level and dendron generation, its polymerization degree ($DP = 28$) revealed a drastic decrease compared with that of **PG1** ($DP = 190$) and **PG2** ($DP = 59$).

To gain more information on the structural features of these rotaxane-branched DPs, small angle neutron scattering (SANS) experiments were then carried out. The scattering profiles of the three samples of **PG1**-**PG3** were first evaluated using the indirect Fourier transformation (IFT) method to obtain a preliminary understanding of the shape of the molecules[67]. As shown in Fig. 4, the scattering curves were converted into their corresponding pair distance distribution functions (PDDF) in real space. Based on the shapes of the PDDFs of these three samples, specific rigid body models were generated for each sample. The scattering profiles of **PG1** and **PG2** were fitted with a rigid cylinder model with cross-sectional radius ($R_{cs}$) values of $2.50 \pm 0.10$ nm and $4.00 \pm 0.10$ nm, respectively, suggesting the successful synthesis of targeted rotaxane-branched DPs. However, due to its smaller polymerization degree, **PG3** was fitted with a sphere model with a $R_g$ value of $3.90 \pm 0.30$ nm (Supplementary Figs. 48–50). In addition, with the help of atomic force microscopy (AFM), the aggregation behaviors of the resultant rotaxane-branched DPs were clearly observed upon deposition on the mica surface. As shown in Supplementary Fig. 51, for rotaxane-branched DPs **PG1** and **PG2**, rod-like aggregates were observed. Notably, for **PG3**, attributed to its relatively small backbone length, a spheroid-like morphology was found.

## Stimuli-responsive rotaxane-branched dendronized polymers with tailored thickness as well as tunable thermal and rheological properties

After confirming the successful synthesis of the targeted rotaxane-branched DPs, their stimuli-responsive properties were then evaluated with regard to the presence of switchable [2]rotaxane units in the attached dendrons. Acetate anions that could bind with the urea moiety were selected as the stimulus. According to the $^1$H NMR titration experiments of **PG1**-**PG3** that were recorded in THF-$d_8$ at 298 K (Supplementary Figs. 54, 56, and Fig. 5), for each [2]rotaxane unit, 5.0 equiv. of tetrabutylammonium acetate (TBAA) was needed to induce the translational motion of the pillar[5]arene macrocycle from the urea station to the alkyl chain moiety in each [2]rotaxane branch. After the addition of TBAA, obvious downfield shifts of the proton signals of the urea moieties ($H_{13}$ and $H_{14}$) as well as remarkable upfield shifts of the proton peaks of the alkyl chain moieties ($H_3$-$H_6$) were observed, indicating the successful switching of the rotaxane-branched DPs from the initial state to a new state with stretched dendrons. Moreover, the further addition of NaPF$_6$ to remove the acetate anions as NaOAc precipitates triggered the pillar[5]arene macrocycle to return to the original urea station, as revealed by the NMR spectra that were almost identical to that of the original state, suggesting the reversible architecture transformation of rotaxane-branched DPs triggered by the addition and removal of acetate anions. Interestingly, as revealed by AFM analysis, upon the addition of acetate anions, **PG1** exhibited

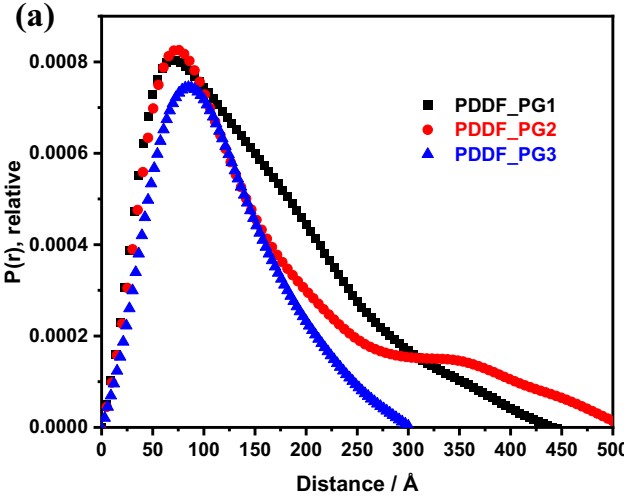

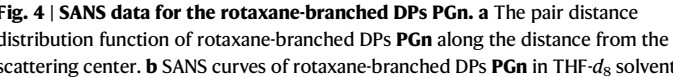

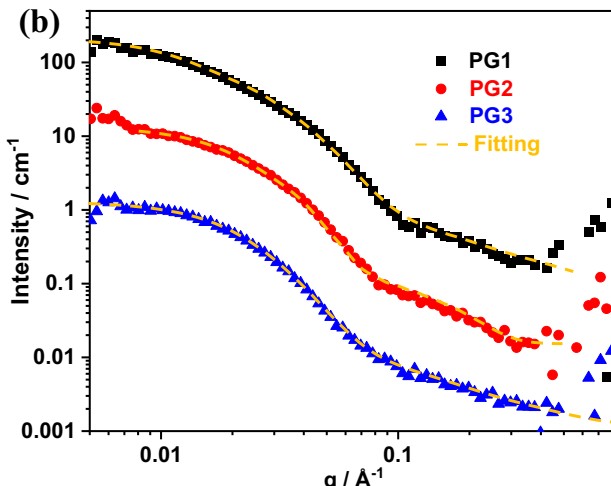

**Fig. 4 | SANS data for the rotaxane-branched DPs PGn. a** The pair distance distribution function of rotaxane-branched DPs **PGn** along the distance from the scattering center. **b** SANS curves of rotaxane-branched DPs **PGn** in THF-$d_8$ solvent and fitting curve based on rigid cylinder + unified model for **PG1** and **PG2** and sphere + unified model for **PG3**.

similar rod-like aggregates, suggesting that the anion-induced thickness modulation did not significantly change its morphology (Supplementary Fig. 52).

Along with the aforementioned reversible thickness modulation process that was further suggested by the structural optimization (Supplementary Fig. 34), the precise regulation of associated parameters, such as local conformations as well as the volume and rigidity of the polymer chains, and the interactions between the individual DPs, was also achieved (Supplementary Fig. 61), which would strongly influence the thermal and rheological properties of the rotaxane-branched DPs. To confirm this, the glass transition temperature ($T_g$) of these rotaxane-branched DPs was obtained from the second differential scanning calorimetry (DSC) heating runs, and the $T_g$ values were 75 °C, 78 °C, 81 °C for **PG1, PG2,** and **PG3,** respectively (Supplementary Fig. 57). The increased $T_g$ of these rotaxane-branched DPs along with generation growths was reasonable since the local mobility was reduced for higher-generation DPs due to the larger and more crowded pendant rotaxane-branched dendrons, which is in line with literature reports[68,69]. More interestingly, after the addition of TBAA, the $T_g$ values of the corresponding rotaxane-branched DPs significantly decreased to 6 °C, 14 °C, and 20 °C for **PG1** + TBAA, **PG2** + TBAA, and **PG3** + TBAA, respectively (Supplementary Fig. 58). These results indicated that, after the addition of TBAA, the back-folding of the backbones became more probable, which could be rationally explained by the increased free chain motions ascribed to the surrounding stretched dendrons. As expected, the further addition of NaPF₆ led to the recovery of the $T_g$ values in a similar trend (43 °C, 70 °C, and 80 °C for **PG1** + TBAA+NaPF₆, **PG2** + TBAA+NaPF₆, and **PG3** + TBAA+NaPF₆, respectively) after one full switching cycle (Supplementary Fig. 59), suggesting the precise modulation of the glass transition temperatures of the rotaxane-branched DPs.

To evaluate the tunable rheological properties of these stimuli-responsive rotaxane-branched DPs, a micronewton shear rheometer mgRheo was employed to perform the rheological measurements with only 2 mg samples or even less[70]. As shown in Fig. 6a, the rheological master curves of rotaxane-branched DPs **PGn** were constructed by shifting the dynamic data to a reference temperature of 150 °C based on the time-temperature superposition (TTS) principle. These master curves revealed behaviors from the terminal region to the rubbery plateau region, and the key difference among these rotaxane-branched DPs was their relaxation times ($\tau$), which were estimated as the inverse frequency of the G′-G″ crossover. As shown in Fig. 6d (black

filled squares), the relaxation times ($\tau$) increased with the generations ($3.2 \times 10^{-2}$ s for **PG1**, $5.3 \times 10^{-2}$ s for **PG2**, and $1.6 \times 10^{-1}$ s for **PG3**), which is possibly due to the enhanced branching degrees along with the increasing generations. Such results were typically observed in branched polymers and other types of thick polymers[71–73]. Upon the addition of TBAA, the side rotaxane branches became relatively more flexible than the initial state. The estimated relaxation times for rotaxane-branched DPs were decreased to $8.7 \times 10^{-4}$ s for **PG1** + TBAA, $3.7 \times 10^{-4}$ s for **PG2** + TBAA, and $1.1 \times 10^{-3}$ s for **PG3** + TBAA (Fig. 6d, red filled circles), which indicated that the relaxation processes of **PGn** + TBAA were remarkably faster than those of **PGn**. This phenomenon was in good agreement with the DSC results. Subsequently, with the further addition of NaPF₆ into the mixture of rotaxane-branched DPs **PGn** and TBAA that could completely drive the pillar[5]arene macrocycle within the rotaxane branches back to the urea moiety, the relaxation processes of **PGn** + TBAA+NaPF₆ were almost returned to the initial state, as revealed by the estimated relaxation times ($5.3 \times 10^{-2}$ s for **PG1** + TBAA+NaPF₆, $9.8 \times 10^{-2}$ s for **PG2** + TBAA+NaPF₆, and $2.8 \times 10^{-1}$ s for **PG3** + TBAA+NaPF₆) (Fig. 6d, blue filled triangles). Notably, as shown in Supplementary Fig. 60, the horizontal shift factors exhibited a Williams-Landel-Ferry (WLF) dependence on temperature for all rotaxane-branched DPs.

## Discussion

In summary, rotaxane-branched DPs were proposed and synthesized through ROMP of macromonomers. More importantly, taking advantaging of the collective motion of each [2]rotaxane branch within the dendrons upon the addition of acetate anions as external stimuli, the controllable modulation of the thickness of resultant rotaxane-branched DPs was successfully realized, which further enabled the regulation of their thermal and rheological properties. By introducing the concept of rotaxane-based molecular switches, this proof-of-concept work not only provides a new switching mechanism for the thickness modulation of DPs, but also greatly expands the toolbox of stimuli-responsive DPs, thus providing an attractive platform for the construction of smart polymeric materials for practical applications.

## Methods

All solvents were dried according to standard procedures and all of them were degassed under N₂ for 30 min before use. All air-sensitive reactions were carried out under an inert N₂ atmosphere. ¹H NMR,

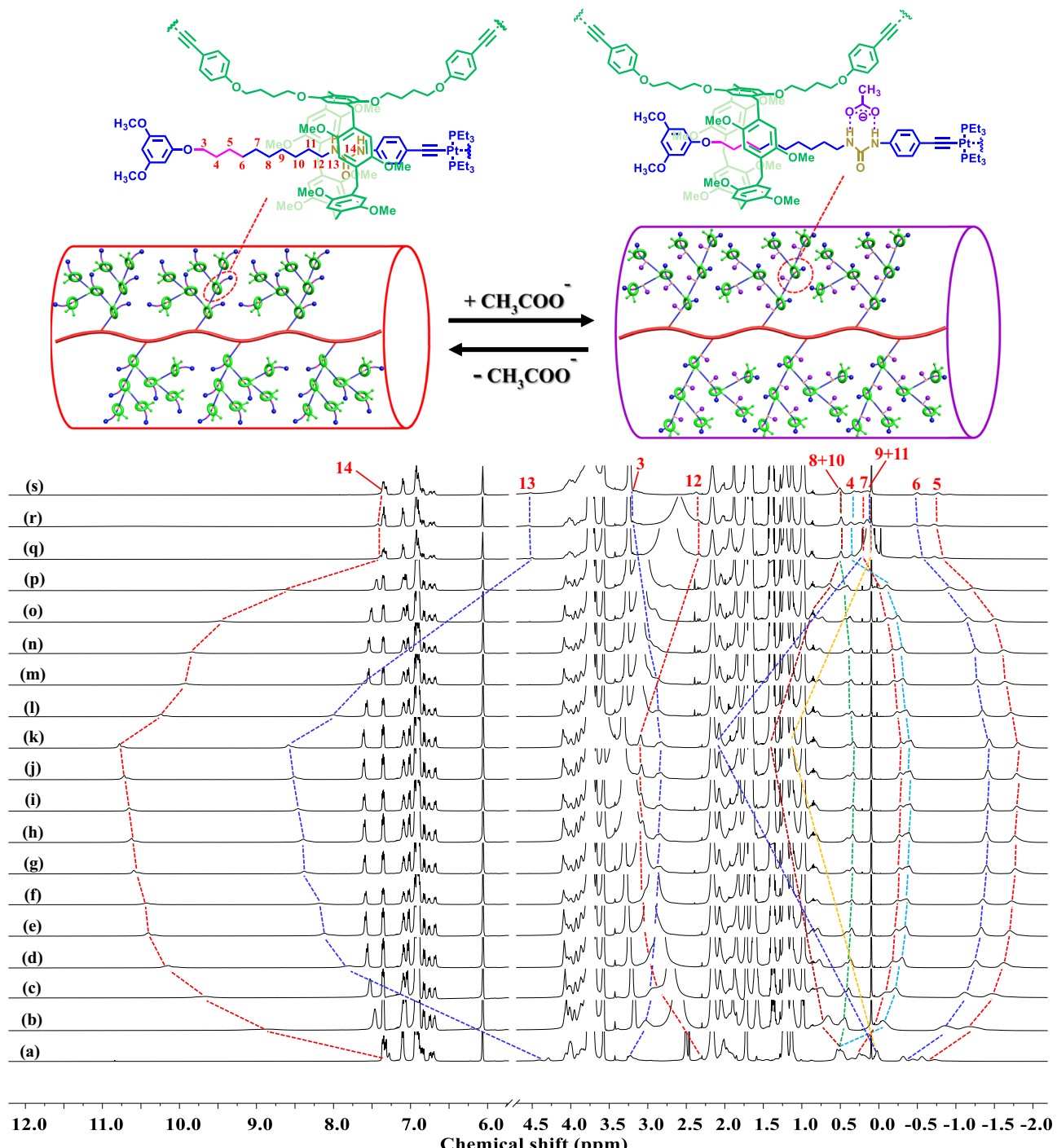

**Fig. 5 | ¹H NMR spectra (THF-$d_8$, 298 K, 500 MHz) of the anion-induced switching behavior of rotaxane-branched DP PG3.** (a) **PG3**; the mixture of **PG3** and TBAA, for each rotaxane unit: (b) TBAA (0.5 equiv.); (c) TBAA (1.0 equiv.); (d) TBAA (1.5 equiv.); (e) TBAA (2.0 equiv.); (f) TBAA (2.5 equiv.); (g) TBAA (3.0 equiv.); (h) TBAA (3.5 equiv.); (i) TBAA (4.0 equiv.); (j) TBAA (4.5 equiv.); (k) TBAA (5.0 equiv.); and the mixture obtained after adding NaPF₆ to the solution in (k), for each rotaxane unit: (l) NaPF₆ (1.0 equiv.); (m) NaPF₆ (2.0 equiv.); (n) NaPF₆ (3.0 equiv.); (o) NaPF₆ (4.0 equiv.); (p) NaPF₆ (5.0 equiv.); (q) NaPF₆ (6.0 equiv.); (r) NaPF₆ (8.0 equiv.); (s) NaPF₆ (10.0 equiv.).

¹³C NMR and ³¹P NMR spectra were recorded on Bruker 300 MHz Spectrometer (¹H: 300 MHz; ³¹P: 122 MHz; ¹³C: 75 MHz), Bruker 400 MHz Spectrometer (¹H: 400 MHz; ³¹P: 162 MHz; ¹³C: 101 MHz), Bruker 500 MHz Spectrometer (¹H: 500 MHz; ³¹P: 202 MHz; ¹³C: 126 MHz) at 298 K. The ¹H and ¹³C NMR chemical shifts are reported relative to residual solvent signals, and ³¹P {¹H} NMR chemical shifts are referenced to an externally unlocked sample of 85% H₃PO₄ ($\delta$ 0.0). The MALDI MS experiments were carried out on a Bruker

UltrafleXtreme MALDI TOF/TOF Mass Spectrometer (Bruker Daltonics, Billerica, MA), equipped with smartbeam-II laser. Electrospray ionization (ESI) mass spectra were recorded with a Waters Synapt G2 mass spectrometer. Gel permeation chromatography (GPC) was carried out at 40 °C using THF as the eluent with a flow rate of 1.0 mL min⁻¹, and the system was calibrated with polystyrene standard. The absolute molecular weights of all the polymers were determined using high-performance size-exclusion

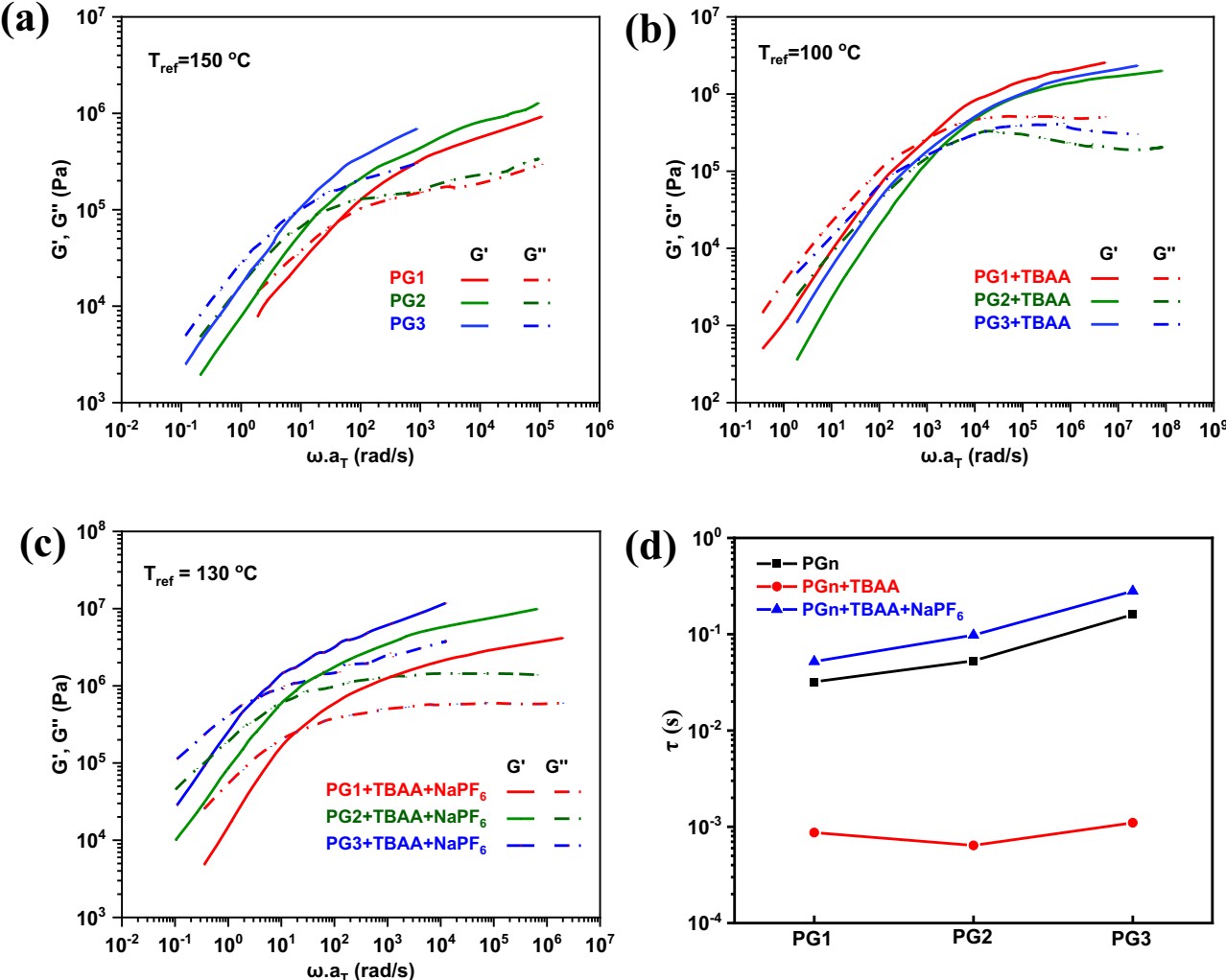

**Fig. 6 | Tunable rheological properties of rotaxane-branched DPs PGn under external stimuli.** Master curves of rotaxane-branched DPs, (**a**) **PGn**, (**b**) **PGn** + TBAA, and (**c**) **PGn** + TBAA + NaPF₆. Relaxation time for rotaxane-branched dendronized polymers **PGn** (black filled squares), **PGn** + TBAA (red filled circles), and **PGn** + TBAA + NaPF₆ (blue filled triangles) (**d**).

chromatography (HPSEC), Viscotek (Viscotek TDAmax) with a differential viscometer (DV), right angle laser-light scattering (RALLS, Viscotek), low-angle laser-light scattering (LALLS, Viscotek), and refractive index (RI) detectors. The column set consisted of a PL 10 mm guard column ($50 \times 7.5$ mm²) followed by one Viscotek T6000 column ($8.0 \times 300$ mm, 10 mm bead size; 104 Å pore size) and one Viscotek T4000 column ($8.0 \times 300$ mm, 6 mm bead size; $1.5 \times 10^3$ Å pore size). A differential scanning calorimeter (DSC) was performed on a Q2000 DSC system in a nitrogen atmosphere. An indium standard was used for temperature and enthalpy calibrations. All the samples were first heated from −40 to 140 °C and held at this temperature for 3 min to eliminate the thermal history, and then, they were cooled to −40 °C and heated again from −40 to 140 °C at a heating or cooling rate of 10 °C min⁻¹. All the AFM images were obtained on a Dimension Fast Scan (Bruker), using ScanAsyst mode under ambient conditions, the samples were prepared by spin casting dilute solutions ($10^{-4}$ mg mL⁻¹) in THF onto freshly cleaved mica for the polymers.

## Data availability
The authors declare that the data supporting this study are available within the paper and its supplementary information file. All other data is available from the authors upon request.

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

## Acknowledgements
H.-B.Y. acknowledges the financial support sponsored by the National Key R&D Program of China (2021YFA1501600), the National Natural Science Foundation of China (92056203), Science and Technology Commission of Shanghai Municipality (21520710200), and the Innovation Program of Shanghai Municipal Education Commission (2019-01-07-00-05-E00012). W.Wang acknowledges the financial support sponsored by the National Natural Science Foundation of China (22001073) and Natural Science Foundation of Shanghai (23ZR1419600). X.-Q.W. acknowledges the financial support sponsored by the National Natural Science Foundation of China (22201077). Y.K. is grateful to the funding of the Youth Innovation Promotion Association, CAS (No.2020010). SANS measurements were performed on the small-angle neutron scattering (SANS) instrument at the China Spallation Neutron Source (CSNS). This work benefited from the use of the SasView application, originally developed under NSF award DMR-0520547. SasView also contains code developed with funding from the European Union's Horizon 2020 research and innovation programme under the SINE2020 project, grant agreement No 654000. Bo Song, Yefei Jiang, and Prof. Lian-Rui Hu (ECNU) are acknowledged for their kind help with the structural simulations.

## Author contributions
H.-B.Y., W.Wang., X.-Q.W., and Y.Z. conceived the project, analyzed the data, and wrote the manuscript. Y.Z. performed the most of experiments. X.-Q.X. synthesized some chemical intermediates. W.Wu carried out the rheological tests under the supervision of G.X.L. H.J. analyzed the SANS data under the supervision of Y.K. W.-J.L., X.-Q.W., and W.-T.X. helped in experiments and data analyses. All authors discussed the results and commented on the manuscript.

## Competing interests
The authors declare no competing interests.
