## [Peer Review File · Nature Communications]

Stimuli-responsive Rotaxane-branched Dendronized Polymers with Tunable Thermal and Rheological PropertiesReviewers' Comments:

Reviewer #1:

Remarks to the Author:

This paper reports the synthesis of a series of polyrotaxane dendrons with a polymerizable group at their focal point of the dendron. These polyrotaxane dendrons were polymerized by the ROMP technique to furnish a series of rotaxane-branched dendronized polymers. These rotaxane-branched dendronized polymers as well as their acetate-triggered rotaxane-switching properties were characterized by NMR spectroscopy, GPC, DSC, AFM, etc.

1. The title was not so specific and seemed like a review paper. More keywords like, platinum-acetylide branch, acetate assembly, thermal and rheology properties, should be considered adding into the title.
2. The introduction part mainly described about the switching and stimuli-responsive features of rod-like dendronized polymers equipped with rotaxane interlocking moieties. What are the implications of the acetate ion assembly and disassembly from the dendronized polymer? It needs more introductory discussion and references for acetate ion sensing, for example.
3. Recent and imperative review articles on rotaxane polymers should be cited, for examples: (a) Chem. Soc. Rev. 2022, 51, 7046-7065; (b) ACS Cent. Sci. 2020, 6, 129-143; (c) Polym. Chem. 2010, 1, 988-1000.
4. In the titration experiment between dendronized polymers and TBA-acetate, what are the reasons of adding 5 equivalents of TBA-acetate for each urea unit? It will be better demonstrating the titration with, for example, 0.5, 1.0, 1.5, 2.0 to 5 equivalents to the dendronized polymers. Perhaps the switching, rigidity and physical properties of the resulting dendronized polymers are different and provided details of the solution switching properties.
5. It would be interesting to characterize also the whereabouts of the tetrabutylammonium (TBA) cation after delivering the acetate to the dendronized polymers.
6. It would be helpful to obtain calculation models for a part of the dendronized polymers before and after the acetate switching, to understand more any non-covalent interactions between each rotaxane-branched dendron on the polymer backbone. These results would correlate and support the claims of the rigidity of the polymers.

Reviewer #2:

Remarks to the Author:

In this manuscript, the authors demonstrated the design and synthesis of a new class of dendronized polymers with rotaxane-branched dendrons attached onto the linear polymer chains. Taking advantage of the switchable feature of [2]rotaxane branches, the controllable thickness modulation of the resultant rotaxane-branched dendronized polymers was successfully realized, thus providing a new strategy for the construction of smart polymeric materials. In particular, they creatively extended the mechanically interlocked macromolecules to a higher level and presented very interesting molecular architectures. Overall, this manuscript describes not only impressive polymer structures but also promising switching properties. I highly recommend its publication after some minor revisions:

1. All synthesis matters described in this manuscript were carried out on a high level, and this is an important accomplishment. As we know, sufficient amount of material is needed for practical uses. How large scale could the targeted rotaxane-branched dendronized polymers be synthesized?

2. The characterization data of the key intermediates MG1-YNE and MG2-YNE should be provided.
3. The switching behavior of the rotaxane-branched dendrons plays the key roles for the thickness modulation, and thus the detailed investigations on the anion-induced switching features of the key monomers are necessary. In particular, additional computational simulations on the anion-induced size modulation are suggested.
4. The modulation of the thermal and rheological properties of rotaxane-branched dendronized polymers through the anion-induced switching of the [2]rotaxane branches is quite impressive, which provides an interesting design strategy for the construction of novel switchable dendronized polymers. The reviewer is wondering how the changes in the polymer structures influences these properties in this study. The additional schematic illustrations might be helpful.

Response to Reviewer 1

This paper reports the synthesis of a series of polyrotaxane dendrons with a polymerizable group at their focal point of the dendron. These polyrotaxane dendrons were polymerized by the ROMP technique to furnish a series of rotaxane-branched dendronized polymers. These rotaxane-branched dendronized polymers as well as their acetate-triggered rotaxane-switching properties were characterized by NMR spectroscopy, GPC, DSC, AFM, etc.

Reply: We greatly appreciate the reviewer's positive comments on our work.

1. The title was not so specific and seemed like a review paper. More keywords like, platinum-acetylide branch, acetate assembly, thermal and rheology properties, should be considered adding into the title.

Reply: Many thanks to reviewer 1 for this excellent suggestion. In the revised manuscript, the title has been modified as "*Stimuli-responsive Rotaxane-branched Dendronized Polymers with Tunable Thermal and Rheological Properties*", in which more keywords have been added.

2. The introduction part mainly described about the switching and stimuli-responsive features of rod-like dendronized polymers equipped with rotaxane interlocking moieties. What are the implications of the acetate ion assembly and disassembly from the dendronized polymer? It needs more introductory discussion and references for acetate ion sensing, for example.

Reply: Based on the reviewer's excellent suggestion, the introductory discussion and references for potential acetate ion sensing have been added in the revised manuscript as follows:

"...In addition, dendronized poly(phenylacetylene)s bearing the second generation lysine dendrons through a urea group has been developed, which could serve as excellent anion receptors for size-selective colorimetric anion sensing such as acetate

anion...”

3. Recent and imperative review articles on rotaxane polymers should be cited, for examples: (a) Chem. Soc. Rev. 2022, 51, 7046-7065; (b) ACS Cent. Sci. 2020, 6, 129-143; (c) Polym. Chem. 2010, 1, 988-1000.

Reply: Many thanks to Reviewer 1 for this excellent suggestion. All these suggested review articles on rotaxane polymers have been cited as references 51-53.

4. In the titration experiment between dendronized polymers and TBA-acetate, what are the reasons of adding 5 equivalents of TBA-acetate for each urea unit? It will be better demonstrating the titration with, for example, 0.5, 1.0, 1.5, 2.0 to 5 equivalents to the dendronized polymers. Perhaps the switching, rigidity and physical properties of the resulting dendronized polymers are different and provided details of the solution switching properties.

Reply: Many thanks for the reviewer’s insightful advice. According to the reviewer’s suggestion, more detailed titration experiments between dendronized polymers and TBA-acetate have been performed. According to the titration results (Figures R1-R3), 5.0 eq. TBA-acetate for each urea unit is needed for the full switching of rotaxane-branched DPs from the initial state to the new state with stretched dendrons, and 10.0 eq. NaPF₆ for each urea unit is further needed to fully switch back to the initial state. In the revised manuscript, additional information on the titration experiments has been added as follows, and the ¹H NMR spectra in Figure 5 has been updated.

“...Acetate anions that could bind with the urea moiety were selected as the stimulus. According to the ¹H NMR titration experiments of PG1-PG3 that were recorded in THF-d₈ at 298 K, for each [2]rotaxane unit, 5.0 equiv. of tetrabutylammonium acetate (TBAA) was needed to induce the translational motion of the pillar[5]arene macrocycle from urea station to the alkyl chain moiety in each [2]rotaxane branch...”

Figure R1. Partial ^1H NMR spectra (THF- d_8 , 298 K, 500 MHz) of anion-induced switching behavior of rotaxane-branched DP **PG1**. (a) **PG1**; the mixture of **PG1** and TBAA, for each rotaxane unit: (b) TBAA (0.5 equiv.); (c) TBAA (1.0 equiv.); (d) TBAA (1.5 equiv.); (e) TBAA (2.0 equiv.); (f) TBAA (2.5 equiv.); (g) TBAA (3.0 equiv.); (h) TBAA (3.5 equiv.); (i) TBAA (4.0 equiv.); (j) TBAA (4.5 equiv.); (k) TBAA (5.0 equiv.); and the mixture obtained after adding NaPF_6 to the solution in (k), for each rotaxane unit: (l) NaPF_6 (1.0 equiv.); (m) NaPF_6 (2.0 equiv.); (n) NaPF_6 (3.0 equiv.); (o) NaPF_6 (4.0 equiv.); (p) NaPF_6 (5.0 equiv.); (q) NaPF_6 (6.0 equiv.); (r) NaPF_6 (8.0 equiv.); (s) NaPF_6 (10.0 equiv.).

Figure R2. Partial ^1H NMR spectra (THF- d_8 , 298 K, 500 MHz) of anion-induced switching behavior of rotaxane-branched DP **PG2**. (a) **PG2**; the mixture of **PG2** and TBAA, for each rotaxane unit: (b) TBAA (0.5 equiv.); (c) TBAA (1.0 equiv.); (d) TBAA (1.5 equiv.); (e) TBAA (2.0 equiv.); (f) TBAA (2.5 equiv.); (g) TBAA (3.0 equiv.); (h) TBAA (3.5 equiv.); (i) TBAA (4.0 equiv.); (j) TBAA (4.5 equiv.); (k) TBAA (5.0 equiv.); and the mixture obtained after adding NaPF_6 to the solution in (k), for each rotaxane unit: (l) NaPF_6 (1.0 equiv.); (m) NaPF_6 (2.0 equiv.); (n) NaPF_6 (3.0 equiv.); (o) NaPF_6 (4.0 equiv.); (p) NaPF_6 (5.0 equiv.); (q) NaPF_6 (6.0 equiv.); (r) NaPF_6 (8.0 equiv.); (s) NaPF_6 (10.0 equiv.).

Figure R3. Partial ^1H NMR spectra (THF- d_8 , 298 K, 500 MHz) of anion-induced switching behavior of rotaxane-branched DP **PG3**. (a) **PG3**; the mixture of **PG3** and TBAA, for each rotaxane unit: (b) TBAA (0.5 equiv.); (c) TBAA (1.0 equiv.); (d) TBAA (1.5 equiv.); (e) TBAA (2.0 equiv.); (f) TBAA (2.5 equiv.); (g) TBAA (3.0 equiv.); (h) TBAA (3.5 equiv.); (i) TBAA (4.0 equiv.); (j) TBAA (4.5 equiv.); (k) TBAA (5.0 equiv.); and the mixture obtained after adding NaPF_6 to the solution in (k), for each rotaxane unit: (l) NaPF_6 (1.0 equiv.); (m) NaPF_6 (2.0 equiv.); (n) NaPF_6 (3.0 equiv.); (o) NaPF_6 (4.0 equiv.); (p) NaPF_6 (5.0 equiv.); (q) NaPF_6 (6.0 equiv.); (r) NaPF_6 (8.0 equiv.); (s) NaPF_6 (10.0 equiv.).

5. It would be interesting to characterize also the whereabouts of the tetrabutylammonium (TBA) cation after delivering the acetate to the dendronized polymers.

Reply: Many thanks for the reviewer's helpful comment. After the addition of TBAA, acetate anions could bind with the urea moiety on the DPs through hydrogen bonding. Since there is no interaction sites with TBA cations on the DPs, the TBA cations might locate around the acetate anions as counteranions according to the following references: *Angew. Chem. Int. Ed.* **2007**, *46*, 6629-6633; *Angew. Chem. Int. Ed.* **2013**, *52*, 10270-

10274; *Langmuir* **2018**, *34*, 6963-6975; *J. Am. Chem. Soc.* **2020**, *142*, 11404-11416; *Angew. Chem. Int. Ed.* **2022**, *61*, e202201793, as shown in the Figure R4.

Figure R4. Cartoon representation of anion-induced thickness modulation of the rotaxane-branched DP **PG3**.

6. It would be helpful to obtain calculation models for a part of the dendronized polymers before and after the acetate switching, to understand more any non-covalent interactions between each rotaxane-branched dendron on the polymer backbone. These results would correlate and support the claims of the rigidity of the polymers.

Reply: According to the reviewer's insightful suggestion, the optimized structures of one repeat unit of rotaxane-branched DP **PG1** before and after the addition of TBAA have been calculated with the aid of the MOPAC2016 program. As shown in Figure R5 (Supplementary Fig. 34 in the updated SI), remarkable stretching of the dendron is observed upon the addition of acetate anions as external stimulus. In the initial state, the distance of N1 (-CONCO-)-Si (TIPS) is 44.72 Å (Figure R5a). Upon the complexation with acetate anion that triggers the pillar[5]arene macrocycles to move from urea moiety to the alkyl chain station, the distance of N1 (-CONCO-)-Si (TIPS) become 54.15 Å (Figure R5b), which indicates the cross-sectional radius (R_{cs}) values of the rotaxane-branched DP **PG1** increase after the addition of acetate anions. These results further support the anion-induced thickness modulation process, therefore affecting the rigidity of the polymers. In the revised manuscript, a brief description has been added as follows:

“...Along with the aforementioned reversible thickness modulation process that was further suggested by the structural optimization (Supplementary Fig. 34), the precise

regulation of associated parameters, such as local conformations as well as the volume and rigidity, of the polymer chains was also achieved, which would strongly influence the thermal and rheological properties of the rotaxane-branched DPs...”

Figure R5. Optimized structures of one repeat unit of rotaxane-branched DP PG1 before (a) and after (b) the addition of TBAA as stimulus with the aid of the MOPAC2016 program. Analytical frequency computations were carried out at the same theoretical level for all stationary points to verify them as intermediate with no imaginary frequency. Structure volume and size variations were analyzed using Multiwfn Software. The visualization of the structures was generated by VMD software.

Response to Reviewer 2

In this manuscript, the authors demonstrated the design and synthesis of a new class of dendronized polymers with rotaxane-branched dendrons attached onto the linear polymer chains. Taking advantage of the switchable feature of [2]rotaxane branches, the controllable thickness modulation of the resultant rotaxane-branched dendronized polymers was successfully realized, thus providing a new strategy for the construction of smart polymeric materials. In particular, they creatively extended the mechanically interlocked macromolecules to a higher level and presented very interesting molecular architectures. Overall, this manuscript describes not only impressive polymer structures but also promising switching properties. I highly recommend its publication after some minor revisions:

Reply: We greatly appreciate the reviewer's positive comments on our work.

1. All synthesis matters described in this manuscript were carried out on a high level, and this is an important accomplishment. As we know, sufficient amount of material is needed for practical uses. How large scale could the targeted rotaxane-branched

dendronized polymers be synthesized?

Reply: In our study, based on the highly efficient ROMP techniques, the targeted rotaxane-branched dendronized polymers could be synthesized on ca. 500 mg scale, which is sufficient for some practical use. Actually, currently we are working on the construction of novel soft actuators based these rotaxane-branched DPs. In the revised manuscript, a brief description on the scale of rotaxane-branched DPs has been added as follows:

“...Notably, all the rotaxane-branched DPs could be synthesized on ca. 500 mg scale, which should be sufficient for practical use...”

2. The characterization data of the key intermediates MG1-YNE and MG2-YNE should be provided.

Reply: According to the reviewer's advice, the characterization data of the key intermediates **MG1-YNE** and **MG2-YNE** have been provided in the updated Supplementary Information (Figures R6-13, Supplementary Figs. 15-18 and Supplementary Figs. 24-27), and the details are shown below:

MG1-YNE: ^1H NMR (400 MHz, CD_2Cl_2 , 298 K) δ 7.43-7.38 (m, 4H), 7.32-7.28 (m, 2H), 7.22-7.14 (m, 4H), 6.98 (d, $J = 3.0$ Hz, 3H), 6.96 (s, 2H), 6.89-6.81 (m, 10H), 6.76-6.72 (m, 2H), 6.28 (t, $J = 1.9$ Hz, 2H), 6.08 (s, 3H), 4.12-3.57 (m, 66H), 3.47-3.41 (m, 2H), 3.22 (m, 2H), 3.04 (d, $J = 2.8$ Hz, 2H), 2.84 (t, $J = 4.4$ Hz, 1H), 2.65 (d, $J = 1.4$ Hz, 2H), 2.22-2.12 (m, 14H), 2.09-1.90 (m, 10H), 1.78-1.70 (m, 3H), 1.50-1.43 (m, 5H), 1.40-1.32 (m, 3H), 1.25-1.20 (m, 25H), 1.05-0.93 (m, 5H), 0.62-0.53 (m, 2H), -0.20 (m, 2H), -1.68 (m, 2H), -1.96 (m, 2H). ^{31}P NMR (162 MHz, CD_2Cl_2 , 298 K): δ 11.52. ^{13}C NMR (101 MHz, CD_2Cl_2 , 298 K): δ 178.30, 162.07, 161.52, 160.04, 159.93, 157.28, 154.43, 151.24, 151.09, 150.72, 150.25, 150.16, 138.19, 133.94, 133.91, 132.16, 131.54, 129.44, 129.17, 129.05, 128.96, 128.70, 119.30, 115.31, 114.93, 114.83, 114.55, 114.39, 114.23, 113.49, 93.77, 92.85, 84.04, 83.91, 76.05, 75.97, 68.68, 68.49, 68.31, 68.24, 57.20, 56.85, 56.73, 55.78, 48.22, 45.64, 43.07, 40.10, 38.87, 31.44, 30.81, 30.47, 30.26, 30.11, 29.84, 29.52, 28.08, 27.12, 26.81, 26.70, 26.49, 26.03, 25.30, 16.81, 8.57. LRMS (MALDI-TOF-MS): Calculated for [**MG1-YNE**+H] $^+$: 2312.1; Found:

2312.2.

Figure R6. ¹H NMR spectrum (CD₂Cl₂, 298 K, 400 MHz) of MG1-YNE.

Figure R7. ³¹P NMR spectrum (CD₂Cl₂, 298 K, 162 MHz) of MG1-YNE.

Figure R8. ^{13}C NMR spectrum (CD_2Cl_2 , 298 K, 101 MHz) of MG1-YNE.

Figure R9. LRMS (MALDI-TOF-MS) spectrum of MG1-YNE. The peak of $m/z = 2312.2$ was observed, which agreed well with the theoretical value of $[\text{MG1-YNE}+\text{H}]^+$ ion ($m/z = 2312.1$).

MG2-YNE: ^1H NMR (400 MHz, CD_2Cl_2 , 298 K) δ 7.44-7.38 (m, 9H), 7.31 (dd, $J = 8.6, 6.6$ Hz, 9H), 7.19 (m, 14H), 7.04-6.94 (m, 19H), 6.91-6.79 (m, 32H), 6.78-6.70 (m, 7H), 6.28 (t, $J = 1.9$ Hz, 2H), 6.10-6.07 (m, 9H), 4.09-3.58 (m, 189H), 3.44 (t, $J = 7.2$ Hz, 1H), 3.22 (t, $J = 1.8$ Hz, 1H), 3.04 (d, $J = 2.6$ Hz, 2H), 2.92 (s, 2H), 2.83-2.76 (d, $J = 4.7$ Hz, 2H), 2.64 (d, $J = 1.4$ Hz, 2H), 2.27-2.16 (m, 44H), 2.10-1.88 (m, 32H), 1.80-1.71 (m, 4H), 1.68-1.60 (m, 6H), 1.31-1.07 (m, 84H), 0.98 (m, 15H), 0.57 (m, 6H), -0.16 (m, 6H), -1.56--1.83 (m, 8H), -1.94 (m, 4H). ^{31}P NMR (162 MHz, CD_2Cl_2 , 298 K): δ 11.50, 11.46. ^{13}C NMR (126 MHz, CD_2Cl_2 , 298 K): δ 178.29, 162.07, 161.51, 160.03, 159.92, 157.27, 154.42, 151.23, 151.07, 151.04, 150.75, 150.72, 150.68, 150.24, 150.15, 138.18, 138.03, 133.94, 133.91, 132.20, 131.53, 129.43, 129.17, 129.05, 128.96, 128.75, 128.72, 128.70, 119.30, 115.30, 114.93, 114.83, 114.54, 114.39, 114.23, 113.49, 93.77, 92.84, 84.03, 83.91, 76.07, 76.00, 68.68, 68.49, 68.31, 68.22, 57.19, 56.85, 56.72, 55.78, 55.68, 48.21, 45.63, 43.06, 40.10, 38.87, 31.44, 30.81, 30.47, 30.26, 30.10, 29.81, 29.52, 28.08, 27.11, 26.99, 26.81, 26.70, 26.48, 26.02, 25.48, 25.30, 16.82, 8.58. LRMS (MALDI-TOF-MS): Calculated for $[\text{MG2-YNE}+\text{H}]^+$: 6211.3; Found: 6211.3.

Figure R10. ^1H NMR spectrum (CD_2Cl_2 , 298 K, 400 MHz) of MG2-YNE.

Figure R11. ^{31}P NMR spectrum (CD_2Cl_2 , 298 K, 162 MHz) of MG2-YNE.

Figure R12. ^{13}C NMR spectrum (CD_2Cl_2 , 298 K, 101 MHz) of MG2-YNE.

Figure R13. LRMS (MALDI-TOF-MS) spectrum of **MG2-YNE**. The peak of $m/z = 6211.3$ was observed, which agreed well with the theoretical value of $[\text{MG2-YNE}+\text{H}]^+$ ion ($m/z = 6211.3$).

3. The switching behavior of the rotaxane-branched dendrons plays the key roles for the thickness modulation, and thus the detailed investigations on the anion-induced switching features of the key monomers are necessary.

Reply: According to the reviewer's excellent suggestion, the detailed investigations on the anion-induced switching features of the key monomers have been provided in the revised manuscript (Figures R14-16, Supplementary Figs. 33, 36 and 38 in the updated Supplementary Information). According to the results, these key monomers revealed similar reversible anion-induced switchable features with that of the corresponding rotaxane-branched DPs.

Figure R14. Partial ^1H NMR spectra ($\text{THF-}d_8$, 298 K, 500 MHz) of anion-induced switching behavior of the first-generation rotaxane-branched dendrimer macromonomer **MG1**. (a) **MG1**; the mixture of **MG1** and TBAA, for each rotaxane unit: (b) TBAA (0.5 equiv.); (c) TBAA (1.0 equiv.); (d) TBAA (1.5 equiv.); (e) TBAA (2.0 equiv.); (f) TBAA (2.5 equiv.); (g) TBAA (3.0 equiv.); (h) TBAA (3.5 equiv.); (i) TBAA (4.0 equiv.); (j) TBAA (4.5 equiv.); (k) TBAA (5.0 equiv.); and the mixture obtained after adding NaPF_6 to the solution in (k), for each rotaxane unit: (l) NaPF_6 (1.0 equiv.); (m) NaPF_6 (2.0 equiv.); (n) NaPF_6 (3.0 equiv.); (o) NaPF_6 (4.0 equiv.); (p) NaPF_6 (5.0 equiv.); (q) NaPF_6 (6.0 equiv.); (r) NaPF_6 (8.0 equiv.); (s) NaPF_6 (10.0 equiv.).

Figure R15. Partial ^1H NMR spectra (THF- d_8 , 298 K, 500 MHz) of anion-induced switching behavior of the first-generation rotaxane-branched dendrimer macromonomer **MG2**. (a) **MG2**; the mixture of **MG2** and TBAA, for each rotaxane unit: (b) TBAA (0.5 equiv.); (c) TBAA (1.0 equiv.); (d) TBAA (1.5 equiv.); (e) TBAA (2.0 equiv.); (f) TBAA (2.5 equiv.); (g) TBAA (3.0 equiv.); (h) TBAA (3.5 equiv.); (i) TBAA (4.0 equiv.); (j) TBAA (4.5 equiv.); (k) TBAA (5.0 equiv.); and the mixture obtained after adding NaPF_6 to the solution in (k), for each rotaxane unit: (l) NaPF_6 (1.0 equiv.); (m) NaPF_6 (2.0 equiv.); (n) NaPF_6 (3.0 equiv.); (o) NaPF_6 (4.0 equiv.); (p) NaPF_6 (5.0 equiv.); (q) NaPF_6 (6.0 equiv.); (r) NaPF_6 (8.0 equiv.); (s) NaPF_6 (10.0 equiv.).

Figure R16. Partial ^1H NMR spectra (THF- d_8 , 298 K, 500 MHz) of anion-induced switching behavior of the first-generation rotaxane-branched dendrimer macromonomer **MG3**. (a) **MG3**; the mixture of **MG3** and TBAA, for each rotaxane unit: (b) TBAA (0.5 equiv.); (c) TBAA (1.0 equiv.); (d) TBAA (1.5 equiv.); (e) TBAA (2.0 equiv.); (f) TBAA (2.5 equiv.); (g) TBAA (3.0 equiv.); (h) TBAA (3.5 equiv.); (i) TBAA (4.0 equiv.); (j) TBAA (4.5 equiv.); (k) TBAA (5.0 equiv.); and the mixture obtained after adding NaPF_6 to the solution in (k), for each rotaxane unit: (l) NaPF_6 (1.0 equiv.); (m) NaPF_6 (2.0 equiv.); (n) NaPF_6 (3.0 equiv.); (o) NaPF_6 (4.0 equiv.); (p) NaPF_6 (5.0 equiv.); (q) NaPF_6 (6.0 equiv.); (r) NaPF_6 (8.0 equiv.); (s) NaPF_6 (10.0 equiv.).

4. In particular, additional computational simulations on the anion-induced size modulation are suggested.

Reply: According to the reviewer's insightful suggestion, the optimized structures of one repeat unit of rotaxane-branched DP **PG1** before and after the addition of TBAA have been calculated with the aid of the MOPAC2016 program. As shown in Figure R17 (Supplementary Fig. 34 in the updated SI), remarkable stretching of the dendron is observed upon the addition of acetate anions as external stimulus. In the initial state,

the distance of N1 (-CONCO-)-Si (TIPS) is 44.72 Å (Figure R5a). Upon the complexation with acetate anion that triggers the pillar[5]arene macrocycles to move from urea moiety to the alkyl chain station, the distance of N1 (-CONCO-)-Si (TIPS) become 54.15 Å (Figure R5b), which indicates the cross-sectional radius (R_{cs}) values of the rotaxane-branched DP **PG1** increase after the addition of acetate anions. These results further support the anion-induced thickness modulation process, therefore affecting the rigidity of the polymers. In the revised manuscript, a brief description has been added as follows:

“...Along with the aforementioned reversible thickness modulation process that was further suggested by the structural optimization (Supplementary Fig. 34), the precise regulation of associated parameters, such as local conformations as well as the volume and rigidity, of the polymer chains was also achieved, which would strongly influence the thermal and rheological properties of the rotaxane-branched DPs...”

Figure R17. Optimized structures of one repeat unit of rotaxane-branched DP **PG1** before (a) and after (b) the addition of TBAA as stimulus with the aid of the MOPAC2016 program. Analytical frequency computations were carried out at the same theoretical level for all stationary points to verify them as intermediate with no imaginary frequency. Structure volume and size variations were analyzed using Multiwfn Software. The visualization of the structures was generated by VMD software.

5. The modulation of the thermal and rheological properties of rotaxane-branched dendronized polymers through the anion-induced switching of the [2]rotaxane branches is quite impressive, which provides an interesting design strategy for the construction of novel switchable dendronized polymers. The reviewer is wondering how the changes in the polymer structures influences these properties in this study. The additional

schematic illustrations might be helpful.

Reply: According to the reviewer's excellent suggestion, the proposed schematic illustration has been added in the updated Supplementary Information as Supplementary Fig. 61. As shown in Figure R18 (Supplementary Fig. 61 in the updated SI), after the addition of TBAA, the movement of pillar[5]arene macrocycles in the rotaxane dendrons from urea moiety to the alkyl chain station leads to the enhanced thickness and flexibility of the side rotaxane dendrons. Along with such switching process, local conformations, the volume and rigidity of the polymer chains, and the interactions between the individual DPs has also been regulated, thus further influencing the thermal and rheological properties of the rotaxane-branched DPs. In the revised manuscript, a brief description on the possible mechanism of the tunable thermal and rheological properties of the rotaxane-branched DPs through the anion-induced switching of the rotaxane branches has been added as follows:

“... Along with the aforementioned reversible thickness modulation process that was further suggested by the structural optimization (Supplementary Fig. 34), the precise regulation of associated parameters, such as local conformations, the volume and rigidity, of the polymer chains, and the interactions between the individual DPs was also achieved (Supplementary Fig. 61), which would strongly influence the thermal and rheological properties of the rotaxane-branched DPs...”

Figure R18. The proposed cartoon representation of tunable thermal and rheological properties of the rotaxane-branched DP PG3 through the anion-induced switching of the rotaxane branches.

Again we greatly appreciate the reviewers' thoughtful suggestions that obviously improved the quality of our manuscript. With these changes and responses, we hope the revised manuscript is now acceptable for publication in *Nature Communications*.

Reviewers' Comments:

Reviewer #1:

Remarks to the Author:

The revised manuscript entitled "Stimuli-responsive Rotaxane-branched Dendronized Polymers with Tunable Thermal and Rheological Properties" provided necessary revisions including a more specific title, detailed NMR titration analysis and supported by initial theoretical calculation. The results are of appropriately high quality to be accepted for publication.

Reviewer #2:

Remarks to the Author:

As the authors have well addressed the comments, the revised manuscript is recommended for publication.

Response to Reviewer 1

The revised manuscript entitled “Stimuli-responsive Rotaxane-branched Dendronized Polymers with Tunable Thermal and Rheological Properties” provided necessary revisions including a more specific title, detailed NMR titration analysis and supported by initial theoretical calculation. The results are of appropriately high quality to be accepted for publication.

Reply: Many thanks for the reviewer's recognition of our revised manuscript.

Response to Reviewer 2

As the authors have well addressed the comments, the revised manuscript is recommended for publication.

Reply: We thank the reviewer for providing a positive comment and allowing publication of this paper in *Nature Communications*.

Again we greatly appreciate the reviewers' thoughtful suggestions that obviously improved the quality of our manuscript. With these changes and responses, we hope the revised manuscript is now acceptable for publication in *Nature Communications*.